# Microbial Composition of Extracted Dental Alveoli in Dogs with Advanced Periodontitis

**DOI:** 10.3390/microorganisms12071455

**Published:** 2024-07-17

**Authors:** Laura Šakarnytė, Raimundas Mockeliūnas, Rita Šiugždinienė, Lina Merkevičienė, Marius Virgailis, Jurgita Dailidavičienė, Žaneta Štreimikytė-Mockeliūnė, Modestas Ruzauskas

**Affiliations:** 1Microbiology and Virology Institute, Lithuanian University of Health Sciences, 44307 Kaunas, Lithuania; laura.sakarnyte@lsmu.lt (L.Š.); raimundas.mockeliunas@lsmu.lt (R.M.); rita.siugzdiniene@lsmu.lt (R.Š.); lina.merkeviciene@lsmu.lt (L.M.); marius.virgailis@lsmu.lt (M.V.); zaneta.streimikyte@lsmu.lt (Ž.Š.-M.); 2Department of Anatomy and Physiology, Veterinary Academy, Lithuanian University of Health Sciences, 44307 Kaunas, Lithuania; jurgita.dailidaviciene@lsmu.lt

**Keywords:** canine, microbiota, bacteria, periodontal disease, traditional bacteriology

## Abstract

Periodontitis is a serious gum infection that damages the soft tissue and destroys the bone supporting the teeth. The aim of the study was to investigate the microbiota using traditional microbiology plating and metagenomic sequencing of extracted tooth alveoli in dogs with severe periodontitis. Isolation of culturable microorganisms was performed as part of bacteriological testing to provide bacteriological diagnosis to veterinary surgeons. Metagenomic sequencing was performed using shotgun sequencing on the Illumina HiSeq system platform. The most prevalent species at sites of periodontal infection detected by metagenomic sequencing were *Porphyromonas gulae*, *Prevotella* spp., *Tannerella forsythia*, *Porphyromonas crevioricanis*, *Porphyromonas cangingivalis*, and *Bacteroides heparinolyticus*. *Pasteurella*, *Streptococcus*, and *Neisseria* were the most frequently isolated culturable bacteria from infected sites detected by traditional microbiologic methods. Metagenomic data revealed that these three genera accounted for only 1.6% of all microbiota at the sites of infection. Antimicrobial resistance patterns of the isolated bacteria included resistance to ampicillin, doxycycline, sulfamethoxazole-trimethoprim, ciprofloxacin, colistin, cefotaxime, and chloramphenicol. Antimicrobial-resistant genes detected using shotgun sequencing also showed resistance to aminoglycosides and macrolides. Dogs with periodontal infections carry bacteria that can cause bite infections in humans as well as multi-resistant isolates. Therefore, treatment and prophylaxis or periodontal disease of dogs is important from a One Health perspective.

## 1. Introduction

One of the most prevalent and severe oral inflammatory diseases in dogs is periodontal disease (PD) [1]. It is caused by an accumulation of bacterial plaque on the teeth. Extensive analyses have focused on the prevalence, severity, and risk factors of canine PD, including poor oral care, diet, behavior, environmental factors, and genetic predisposition [2]. However, the causative pathogens remain controversial [3,4]. PD is currently recognized as a syndromic diagnosis rather than a distinct medical condition. It covers a spectrum of specific diagnoses, including gingivitis and periodontitis, with the term used to describe the presence of at least one of these conditions [5,6,7]. Existing data demonstrate that PD affects 44.0% to 63.6% of dogs [8,9].

Despite the high prevalence and significant impact of PD on animal health, there is limited published evidence on its epidemiology [10]. This may be partially because many cases of PD are managed completely in the primary care setting [11]. Moreover, according to published data, most causative agents of PD are Gram-negative anaerobic rods [12] which are difficult to culture. Therefore, the identification of aetiological agents in traditional bacteriology laboratories is limited [13]. 

Periodontitis is a more severe form of infection than gingivitis and occurs in deeper and more restricted dental areas, including the tooth apex and alveolus. Periodontitis develops when bacteria penetrate deeply into the tissues and surrounding periodontium [14]. In a few studies, *Treponema denticola* was considered the bacterial species most commonly associated with PD [15,16]. Kačirova et al. recently found that the same bacterial species were prevalent in both healthy dogs and dogs with PD. These bacteria included *Treponema forsythia*, *Porphyromonas gulae, Treponema denticola*, and *Treponema putidum*. *Treponema putidum* was positively correlated with both *Porphyromonas gulae* and *Treponema forsythia*, suggesting that it may be involved in the development of PD [4]. Other data suggest that PD and its pathogenesis are strongly associated with *Porphyromonas* spp., *Tannerella forsythia*, and *Campylobacter rectus* [17,18,19,20,21]. The wide range of data on various microorganisms implicated in PD demonstrates that the aetiological agents may vary based on geographic area, breed, investigation techniques, or the extensive area surrounding the teeth where both pathogenic and normal microbiota coexist. There is also a lack of data regarding how different strains of the same species within oral microbiomes can vary in pathogenicity, largely because the serotyping of oral microorganisms has been poorly investigated. The use of different methods (such as traditional bacteriology, polymerase chain reaction, and sequencing) for the detection of aetiological agents also contributes to the discrepancies among studies.

Although novel methods such as metagenomic sequencing enable active exploration of the microbiomes in animals, data regarding the microbial taxonomy in dogs with PD are usually restricted to the higher taxa without species determination. For example, Santibáñez et al. determined that PD was associated with a significant increase in Bacteroidetes and reductions in Actinobacteria and Proteobacteria compared with healthy dogs [22]. However, the obtained data lacked information regarding the increasing abundance of particular species, which could vary in pathogenicity and importance in the etiology of the disease. Similarly, Watanabe et al. recently detected *Porphyromonas, Fusobacterium*, *Moraxella*, *Neisseria*, *Pasteurella*, *Actinomyces*, and some other bacterial genera without more detailed data at the species level [23]. These same genera have also been detected in healthy dogs [24,25,26], complicating our understanding of the bacteria involved in the pathogenesis of PD and particularly of periodontitis. Moreover, most previous studies on the oral microbiota of dogs were based on 16S rRNA gene amplicon sequencing, in which a small region of a ribosomal sequence is amplified and sequenced [25]. For deeper insight, metagenomic shotgun sequencing (MGS) can be performed to investigate any part of the genome and provide more information about the microbial community with higher taxonomic resolution than 16S rRNA gene sequencing [25].

Antimicrobial resistance is a serious problem both in human and veterinary medicine. Since pet owners cannot escape close contact with their dogs, this problem requires a One Health approach, and monitoring of antimicrobial resistance in dogs should be performed [27]. Existing data show that antimicrobial-resistant bacteria can spread from dogs to their owners [28].

The aim of the study was to investigate the microbiota using traditional microbiology plating and metagenomic sequencing of extracted tooth alveoli in dogs with severe periodontitis. Antimicrobial resistance patterns were also investigated in isolated bacteria, as well as the genes encoding antimicrobial resistance. 

## 2. Materials and Methods

### 2.1. Animals Involved in the Study

All animals were patients at a veterinary clinic located in central Lithuania. According to the treatment history and questionnaire data, no antibiotics had been used for the dogs during the previous 48 months. No oral hygiene practices (e.g., teeth brushing) were applied to the tested dogs. The general health status was evaluated together with morphological and biochemical blood analyses according to the standard procedures of veterinary inspection. The dogs’ oral cavities were examined by a veterinary odontologist under general anesthesia using inhalant anesthesia protocols and sufficient pain control. In cases of suspected periodontitis, X-ray examinations were performed to evaluate the disease stage. Only dogs that had signs of advanced periodontitis (PD3–PD4), had periodontal pockets of at least 6 mm, had not been treated for at least 6 months, and had no home dental care were initially selected for further analysis. Dogs that required radical treatment (e.g., dental extraction) were finally selected according to the above-mentioned inclusion criteria, ensuring variety in age, sex, feeding type, neutering status, and breed. Dogs less than 1 year old were excluded from the experiment. In total, 30 dogs were selected. In all cases, 109 and/or 110 teeth were extracted, and a radiological examination confirmed bone loss in >50% of cases. The detailed characteristics of the dogs with PD involved in this study are presented in Table 1.

Prior to the commencement of the study, written informed consent was obtained from the owner of each dog. All procedures were performed according to the applicable international, national, and institutional guidelines for the care and use of animals. Ethics approval was obtained from the Bioethics Center, Lithuanian University of Health Sciences, Kaunas, Lithuania (approval number 2024-BEC3-T-005). All animals were from Kaunas County, located in the central part of Lithuania. 

### 2.2. Preparation of Samples

Immediately after the extraction of teeth, two small sterile cotton swabs (Transwab; Medical Wire and Equipment Co., Ltd., Corsham, UK) were inserted into the dental alveoli. One swab was used for plating, and the other swab was stirred and suspended in a DNA/RNA Shield Collection Tube (Zymo Research, Irvine, CA, USA). Samples from each individual dog were obtained from a single alveolus, even when more than one tooth was extracted during the procedure. Thereafter, a pooled sample for next-generation sequencing was prepared by mixing equal parts of all alveolar solutions from the extracted teeth. The sample was placed into a 2 mL cryogenic tube and stored at −80 °C until DNA extraction and sequencing.

### 2.3. Plating and Identification of Culturable Bacteria

The samples from the periodontal alveoli were plated on universal media (Columbia agar with 5% horse blood and fastidious anaerobe agar with 5% horse blood; Liofilchem, Roseto degli Abruzzi, Teramo, Italy). The plates were incubated at 37 °C both in ambient air and anaerobically for up to 72 h. To obtain pure cultures, up to three most prevalent but different types of colonies were recultivated. Thereafter, DNA was extracted, and bacterial identification was performed with Sanger sequencing of 16S rRNA using the universal primers 27F and 515R as described previously [29].

### 2.4. Susceptibility Testing of Culturable Bacteria

Susceptibility testing of the culturable species with existing interpretative clinical breakpoints criteria (*Enterobacteriaceae*, *Pasteurella*, *Neisseria*, *Streptococcus*, and *Staphylococcus*) was performed using Mueller Hinton Agar supplemented with 5% sheep blood (Liofilchem, Italy). Minimum inhibitory concentrations were determined with E-test strips (Liofilchem, Italy). The following antimicrobials were used: ampicillin, doxycycline, sulfamethoxazole-trimethoprim, gentamicin, erythromycin, chloramphenicol, ciprofloxacin, colistin, imipenem, and cefotaxime. Clinical breakpoints were used as interpretative criteria as recommended by the EUCAST [30]. 

### 2.5. Metagenomic Sequencing

DNA isolation, quality control, library preparation, and MGS were performed with a ZymoBIOMICS Targeted Sequencing Service for Microbiome Analysis (Zymo Research). DNA was extracted using a ZymoBIOMICS DNA Microprep Kit (Zymo Research). Libraries were prepared using a Nextera DNA Flex Library Prep Kit (Illumina, San Diego, CA, USA) with up to 100 ng DNA input in accordance with the manufacturer’s instructions using internal dual-index 8 bp barcodes with Nextera adapters (Illumina). Thereafter, the libraries were quantified with TapeStation (Agilent Technologies, Santa Clara, CA, USA) and pooled in equal abundance. The final pool was quantified using a quantitative polymerase chain reaction. The final library was sequenced on an Illumina HiSeq system (Illumina). 

### 2.6. Bioinformatics and Data Analysis

The Trimmomatic-0.33 software tool was used to trim raw sequence reads with the aim of removing low-quality fractions and adapters. Reads of <70 bp were removed. The microbial composition was profiled as described previously [31]. Microorganisms were identified using the full Genome Taxonomy Database (R07-RS207). Sequences were deposited in the National Center for Biotechnology Information database (access number PRJNA1099262).

Antimicrobial resistance gene identification was performed with the DIAMOND sequence aligner [32]. Reads with less than five DNA reads of the same gene were omitted to escape possible data bias. 

All taxonomic ranks of microorganisms were counted as percentage amounts. The numbers of culturable bacterial species obtained using the plating method were evaluated by comparing their DNA reads with all the bacterial reads obtained by metagenomic sequencing.

## 3. Results

The culturable microorganisms isolated from the alveoli of extracted teeth of dogs using the plating technique are presented in Table 2.

Among the 47 isolates selected as predominant culturable microorganisms, 15 different genera were identified. The most prevalent were *Pasteurella*, *Neisseria*, and *Streptococcus*. Antimicrobial resistance patterns of the isolates included resistance to ampicillin, sulfamethoxazole-trimethoprim, chloramphenicol, ciprofloxacin, colistin, gentamicin, and doxycycline. No isolates resistant to carbapenems, aminoglycosides, and macrolides were detected. However, multi-resistant isolates (resistant to at least three different classes of antimicrobials) were detected. Those isolates included *Neisseria weaveri*, *Streptococcus canis*, and *Proteus mirabilis*.

Metagenomic sequencing from dogs with PD detected 15 different taxa at the phylum level. The most prevalent phylum was Bacteroidota (61.5%), followed by Actinomycetota (10.8%) and Firmcutes A (9.1%; Figure 1).

In total, 54 genera of bacteria with a prevalence of at least 0.1% and one phylum of the domain Archaea (Methanobacteriota) were detected using MGS (Appendix A). The most prevalent genera were *Porphyromonas* (31.2%), *Prevotella*, (9.8%), *Bacteroides* (8.9%), *Tannerella* (5.6%), *Peptostreptococcus* (4.3%), and *Porphyromonas* (4.3%; Figure 2). Although *Pasteurella* was the most frequently detected genus from culturable isolates, the proportion of this genus among all microbiota detected by metagenomic sequencing was only 1.0%. The DNA content of the other most frequently detected culturable bacteria, *Neisseria* and *Streptococcus*, accounted for 0.6% of all detected microorganisms for each genus (Appendix A).

In total, 113 microbial species were detected from the dental alveoli of dogs with periodontitis (Appendix A), among which 21 species accounted for ≥1% of the total amount of bacteria (Figure 3). 

The most prevalent species detected by MGS was *Porphyromonas gulae*, with a prevalence of 20.5%. The other most frequently detected species in dental alveoli were *Prevotella* sp. 003932845 (8.4%), *Bacteroides pyogenes* (5.8%), *Tannerella forsythia* (5.3%), *Porphyromonas crevioricanis* (4.4%), *Porphyromonas gingivicanis* (2.6%), *Porphyromonas cangingivalis* (2.6%), and *Bacteroides heparinolyticus* (2.2%).

The genes encoding antimicrobial resistance in the pooled samples of dogs determined using shotgun sequencing are presented in Table 3. The most frequently detected genes were associated with the resistance to tetracyclines and less frequently with macrolides, β-lactams, aminoglycosides, and colistin.

## 4. Discussion

PD affects the periodontium, leading to gingivitis and/or periodontitis of varying severity [33]. Severe periodontitis (stages 3 and 4) is usually untreatable without radical measures; therefore, extraction of teeth is required to restore the general health status and welfare of the dog. In this study, we compared the results of microbiological examinations of alveoli from extracted teeth of dogs with severe periodontitis using both the plating technique (traditional bacteriology) and metagenomic sequencing to assess the differences between the two methods. Most previous studies of PD-associated infectious agents in dogs used either traditional or molecular methods to identify bacteria associated with PD. The following bacterial genera involved in the later stage of PD have been reported by authors studying PD using culturable methods: *Veillonella*, *Bacterioides*, *Prevotella* [34], *Fusobacterium* [35,36], and *Porphyromonas* [34,36,37]. Other authors have used specific primers to detect specific species associated with periodontitis, such as *Porphyromonas gulae*, *Tannerella forsythia*, or *Treponema denticola* [4,38]. In most studies, the plaque from teeth or samples from periodontal pockets was analyzed. Our study is novel in that both traditional and metagenomic methods were used to identify bacteria from the same site of infection, namely the alveoli, immediately after tooth extraction; this reduced the possibility of swab contamination by the external microbiota (which is abundant in the mouth). The data demonstrated that the most prevalent culturable bacteria were *Pasteurella*, *Neisseria*, and *Streptococcus*. Four different *Pasteurella* species (*P. multocida*, *P. stomatis*, *P. canis*, and *P. dagmatis*) were detected in the alveoli of extracted teeth, which may suggest their importance in the pathogenesis of canine PD. *Pasteurella* is an obligative pathogen in different animal species, but only *Pasteurella multocida* is predominant in most domesticated species of vertebrate animals. Some data have shown that in dogs, *Pasteurella* is associated with PD and advanced periodontitis, which can progress to more severe diseases such as meningoencephalitis or endocarditis [39,40]. *Neisseria* also shows variety at the species level; four species were identified in this study, namely *N. animaloris*, *N. zoodegmatis*, *N. weaveri*, and *N. dumasiana*. All these species are known to be prevalent in the oral microbiome of dogs, and they can cause infections after dog or cat bites [41,42,43]. Although this proves their pathogenicity, they still are considered part of the normal canine oral microbiota [24,26,44]. *Streptococcus* is also abundant in the oral microbiomes of animals. We isolated three species of *Streptococcus*: *S. canis*, *S. minor*, and *S. fryi*. Both *S. canis* and *S. minor* are known to be part of the canine oral microbiome but also are considered potential human pathogens associated with bite infections [45,46]. Data regarding *S. fryi* are scarce; this species was first isolated from dogs by [47]. Other culturable bacteria isolated from the extracted tooth alveoli of dogs include a few very well-known species of Enterobacteriaceae, *Staphylococcus pseudintermedius*, and some anaerobes frequently found in the oral canine microbiome, such as *Porphyromonas macacae*, *Bacteroides pyogenes*, and *Fusobacterium polymorphum*.

Metagenomic analysis of samples from the infected sites of extracted teeth demonstrated the highest prevalence of the phylum Bacteroidota (61.5%) among all 15 phyla. The clinical significance of Bacteroidota in carnivores is poorly understood. However, this phylum is known to be a core aspect of the human microbiome, playing roles in dietary glycan foraging, beneficial metabolite production, and immunity [48]. Notably, Bacteroidota contains some genera, such as *Bacteroides*, *Prevotella*, and *Porphyromonas*, that are well known to cause oral infections in both humans and dogs. At the genus level, the most prevalent genus is *Porphyromonas*, which accounts for >30% of the entire microbial composition. Although this genus, together with *Tannerella*, was previously reported to be predominant in dogs with PD [22], high numbers of these bacteria were also recently detected in the oral microbiome of healthy dogs [49]. This demonstrates that evaluating the prevalence of microorganisms at the genus level is not always specific enough, necessitating analysis at the species level or even strain/serotype level. The other most prevalent genera in the alveoli of extracted teeth were *Prevotella*, *Bacteroides*, *Tannerella*, and *Peptostreptococcus*. These genera of bacteria were previously described as those whose numbers increased in cases of PD or were mentioned as causative agents of PD together with several culturable bacteria, such as *Neisseria* sp., *Corynebacterium* sp., *Pasteurella* sp., and *Moraxella* sp. [22,50,51]. 

The oral microbiome of dogs is known to have the potential to cause bite infections in humans. Bacteria associated with bite infections detected in our study included *Bacteroides pyogenes* [52], *Streptococcus canis*, *S. minor* [45,46], and different *Neisseria* species [41,42,43]. The detection of this and other bacteria known to cause infections in humans after bites from dogs with periodontitis underscores the high risk of humans developing serious wound infections after being bitten by dogs with periodontal infections, especially when compared with healthy dogs. For this reason, hygienic measures and treatment of PD are very important not only for the health of dogs but also for the safety of their owners, family members, and community. Our investigations demonstrated a high prevalence of the latest PD stages in dog populations, meaning that dental hygiene in dogs is often neglected and that owners should pay attention to the prophylactic measures against this disease. According to other studies, tooth brushing, the application of different oral products, such as chemical plaque reduction products, enzymes, or derived herbal products, and the usage of specific diet and chew toys can reduce the risk of the development of PD in dogs [1].

Antimicrobial resistance data demonstrated that bacteria isolated from dogs with periodontitis were resistant to some classes of critically important antibiotics to humans. These antibiotics included third-generation cefalosporins, fluoroquinolones, and colistin. Phenotypical resistance detected in culturable Comparison of microbiota from the sites of infections with the microbiome of healthy dogs can help to understand the differences between pathogenic and normal oral canine microorganisms. Although the data about the oral microbiome of healthy dogs are still scarce, some studies revealed the normal composition of bacteria in healthy individuals. For example, Lisjak et al. [25] identified 67 bacterial species as core oral microbiota in dogs. The most prevalent species detected in their study were *Porphyromonas cangingivalis*, *P. gulae*, *P. canorus*, *P. gingivicanis*, *Conchiformibius steedae*, *Nesseria weaveri*, *Frederiksenia canicola*, *Capnocytophaga cynodegmi*, *C. canimorsus*, *C. canis*, and *Bergeyella zoohelcum*, which consisted of ≥1% of the oral microbiome of dogs. The other study has been performed recently by Šakarnytė with colleagues [49] in the same geographical area and using the same methods as those used in this study. In dogs without signs of periodontitis, they found *Porphyromonas guleae*, *P. cangingivalis*, *P. canoris*, *Pauljensenia canis*, *P. gingivicanis*, *Corynebacterium canis*, *C. freiburgense*, *Neisseria dumasiana*, *Actinomyces wessii*, and *Lampropedia* spp. as the most prevalent species. When comparing those species with bacteria isolated from extracted teeth alveoli, it is obvious that *Prevotella* sp. 003932845, *Bacteroides pyogenes*, *Tannerella forsythia*, *Aminobacteriaceae* sp., *Porphyromonas crevioricanis*, *Peptostreptococcus canis*, and *Porphyromonas gingivalis* had much higher percentage amounts in dogs with PD than in healthy individuals or those bacteria in healthy dogs were not detected at all. Therefore, these species can be associated with the pathogenesis of PD. Bacteria were overall similar to the genes encoding resistance detected by MGS. However, in culturable bacteria, we found no resistance to macrolides and aminoglycosides, which were genotypically detected in the total DNA of the samples. This can be explained by the fact that only a small part of the microbiomes was analyzed with traditional susceptibility testing and that molecular methods can provide more detailed information about the resistome in the infected area. Multi-resistant isolates were detected in the dogs during this study. Such resistant bacteria can be the reason for antimicrobial treatment failure in dogs and also pose a risk to other animals in contact with dog owners. These results are similar to the results obtained by some other authors, who found multi-resistant isolates in dogs that pose a risk to humans [27,53].

## 5. Conclusions

High abundance of *Prevotella* sp. 003932845, *Tannerella forsythia*, *Porphyromonas crevioricanis*, *Porphyromonas cangingivalis*, *Bacteroides heparinolyticus*, and *Bacteroides pyogenes* in dental alveoli of dogs with the third and fourth stage periodontal infections suggest that the above-mentioned bacteria can be potential causative agents of severe periodontitis. The most prevalent species at sites of periodontitis among culturable bacteria (*Pasteurella*, *Streptococcus*, and *Corynebacterium*) accounted for only 1.6% of all bacteria detected using metagenomics. It indicates that bacteriological diagnosis is not always able to provide a real-world view of the microbial variety at sites of infection. This could lead to an incorrect bacteriological diagnosis of aetiological agents and subsequent inappropriate selection of antimicrobials for treatment. Antimicrobial treatment, if required, should be performed only after susceptibility testing, as some of the canine isolates from dogs with periodontitis are resistant to different classes of antibiotics, including critically important ones. Dogs with periodontal infections carry bacteria that can cause bite infections in humans as well as multi-resistant isolates. Therefore, treatment and prophylaxis of periodontal disease in dogs is not only important for animal health but also from the perspective of a One Health approach.

## Figures and Tables

**Figure 1 microorganisms-12-01455-f001:**
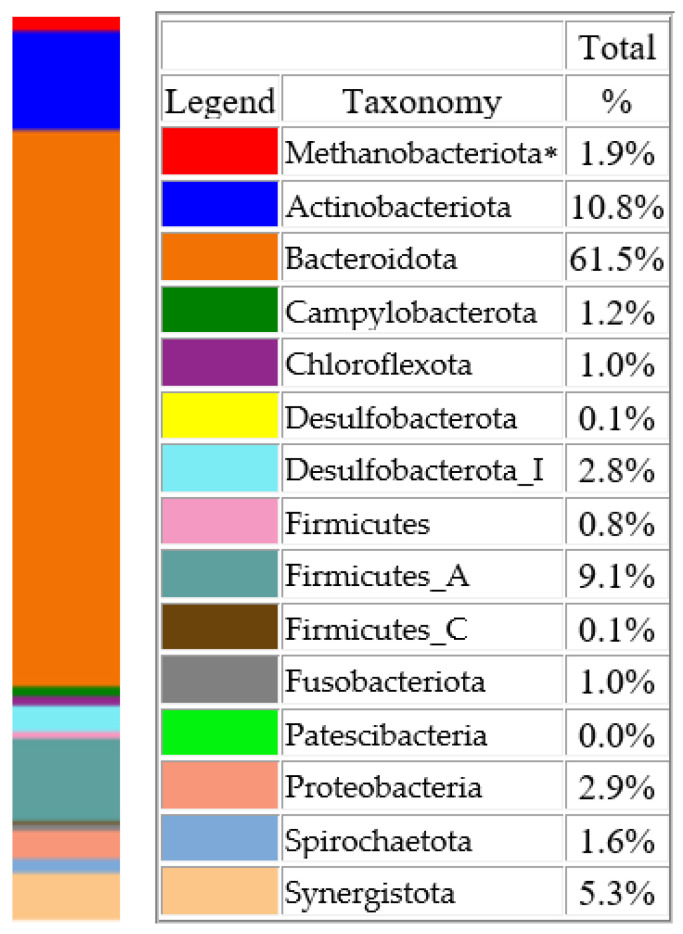
Microorganisms at a phylum level detected using MGS in dental alveoli of dogs with periodontal disease. ***** Euryarchaeota.

**Figure 2 microorganisms-12-01455-f002:**
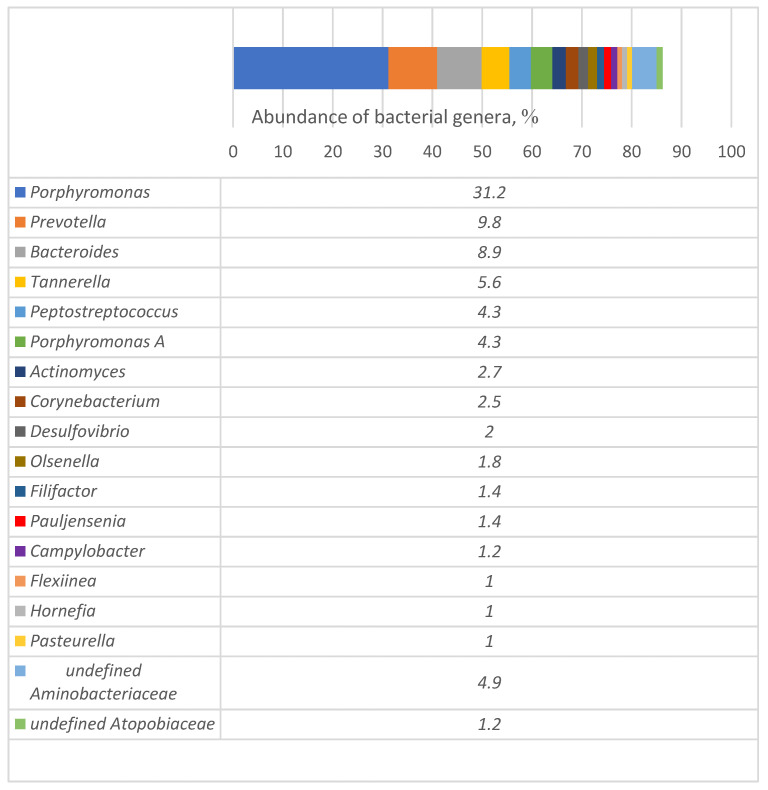
Most prevalent genera of bacteria detected in alveoli of extracted teeth from dogs with periodontal disease. Only genera with a prevalence of at least 1% among the total reads are presented.

**Figure 3 microorganisms-12-01455-f003:**
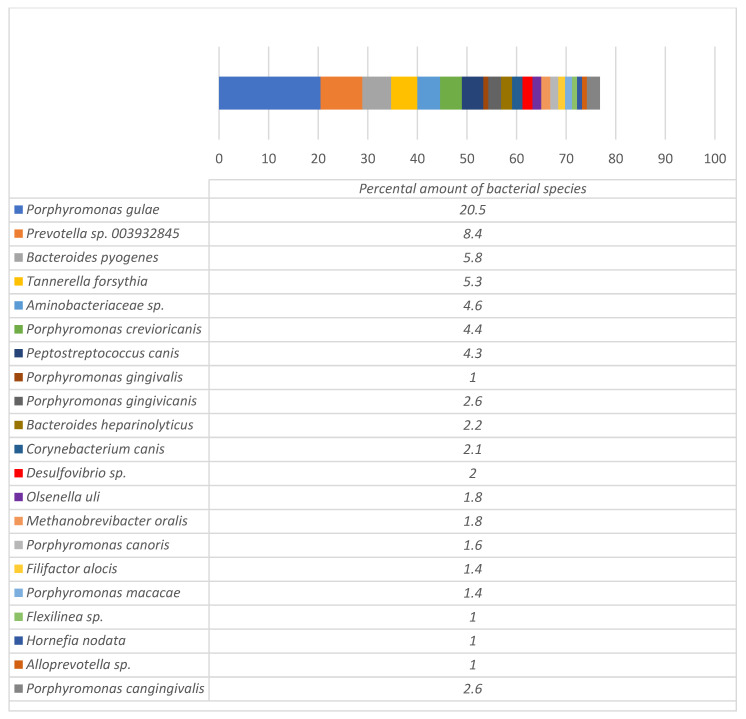
The most prevalent species detected in dental pockets of dogs with severe periodontitis.

**Table 1 microorganisms-12-01455-t001:** Dogs with advanced periodontal disease in the present study.

Dog Number	Age, Years	Sex ^1^	Weight, kg	Nutrition Type	PD Level ^2^	CI Level ^3^	Pedigree	Breed	Neutered
1	10	M	3.4	Mix	PD4	3	No	Yorkshire terrier	Yes
2	11	F	14.6	Mix	PD4	3	No	Mixed breed	Yes
3	7	F	18.4	Raw	PD3	3	Yes	Mittelschnauzer	No
4	12	M	28.0	Mix	PD3	3	No	Mixed breed	Yes
5	10	M	6.3	Mix	PD3	3	No	Mixed breed	Yes
6	10	F	9.3	Mix	PD3	3	No	Mixed breed	Yes
7	12	M	4.0	Dry	PD3	3	No	Yorkshire terrier	Yes
8	6	M	14.0	Mix	PD3	2	Yes	French bulldog	No
9	9	M	37.0	Mix	PD3	3	Yes	Bobtail	No
10	7	M	10.0	Dry	PD3	2	Yes	West Highland terrier	No
11	7	M	13.0	Dry	PD3	3	Yes	French bulldog	Yes
12	12	F	12.0	Mix	PD3	3	No	Beagle	Yes
13	6	F	12.3	Dry	PD3	3	No	French bulldog	No
14	10	M	5.5	Mix	PD3	2	No	Mixed breed	No
15	10	M	1.7	Wet	PD4	3	Yes	Russian toy terrier	Yes
16	12	M	2.1	Wet	PD4	3	Yes	Russian toy terrier	No
17	15	F	2.5	Wet	PD4	3	No	Russian toy terrier	Yes
18	9	M	11.0	Wet	PD4	3	No	Mixed breed	Yes
19	7	F	12.0	Wet	PD3	3	No	Mixed breed	Yes
20	7	F	3.0	Mix	PD4	3	No	Pomeranian	Yes
21	5	F	4.2	Mix	PD3	3	No	Russian toy terrier	No
22	7	M	5.2	Mix	PD4	3	No	Mixed breed	Yes
23	9	F	4.0	Dry	PD3	3	No	Maltese	No
24	13	F	8.5	Mix	PD4	3	Yes	French bulldog	No
25	15	M	11.0	Wet	PD4	3	Yes	English cocker spaniel	Yes
26	10	M	5.3	Mix	PD4	3	Yes	Pomeranian	Yes
27	10	F	7.4	Mix	PD3	2	No	Mixed breed	Yes
28	5	F	2.4	Mix	PD3	3	Yes	Pomeranian	Yes
29	12	F	4.2	Dry	PD3	3	Yes	Yorkshire terrier	Yes
30	9	F	5.8	Mix	PD3	3	Yes	Shih Tzu	No

^1^ M, male; F, female; ^2^ PD, periodontal disease; ^3^ CI, clinical attachment.

**Table 2 microorganisms-12-01455-t002:** Culturable bacterial species identified and their phenotypic antimicrobial resistance patterns.

Isolate Number	PD Stage *	Species Identified Using NCBI Database **	Antimicrobial Resistance Profile
1	PD4	*Actinomyces* sp. canine oral taxon ^n^	
2	PD4	*Neisseria animaloris*	Cip
3	PD4	*Neisseria weaveri*	Cip, Chlor, Amp
4	PD3	*Serratia* sp.	Ctx
5	PD3	*Pasteurellaceae* sp.	Cip
6	PD3	*Neisseria dumasiana*	-
7	PD3	*Pasteurella canis*	-
8	PD3	*Staphylococcus pseudintermedius*	-
9	PD3	*Pasteurellaceae* sp.	-
10	PD3	*Streptococcus canis*	Doxyy, Sxt, Amp
11	PD3	*Pasteurella stomatis*	-
12	PD3	*Leucobacter* sp. canine oral taxon ^n^	
13	PD3	*Proteus mirabilis*	Cip, Sxt, Chlor, Amp
14	PD3	*Glaesserella parasuis* ^n^	
15	PD3	*Pasteurella canis*	-
16.	PD3	*Pasteurellaceae* sp.	-
17	PD3	*Pasteurella multocida*	-
18	PD3	*Pasteurella multocida*	-
19	PD3	*Neisseria animaloris*	Cip
20	PD4	*Neisseria animaloris*	-
21	PD4	*Neisseria animaloris*	Cip
22	PD4	*Streptococcus minor*	-
23	PD3	*Neisseria zoodegmatis*	-
24	PD3	*Franklinella schreckenbergeri* ^n^	
25	PD3	*Neisseria animaloris*	-
26	PD3	*Pasteurella dagmatis*	-
27	PD4	*Neisseria zoodegmatis*	-
28	PD4	*Streptococcus minor*	-
29	PD4	*Pasteurella stomatis*	-
30	PD4	*Pasteurella multocida*	Cip, Sxt
31	PD4	*Streptococcus fryi*	-
32	PD4	*Neisseria animaloris*	-
33	PD3	*Pasteurella canis*	-
34	PD3	*Pasteurella dagmatis*	-
35	PD3	*Rahnella aquatilis*	-
37	PD3	*Proteus mirabilis*	Amp, Cip
38	PD3	*Peptostreptococcus* sp. ^n^	
39	PD3	*Proteus mirabilis*	Amp, Cip, Col, Ctx
40	PD3	*Glaesserella parasuis* ^n^	
41	PD3	*Streptococcus canis*	Doxy
42	PD3	*Streptococcus canis*	-
43	PD3	*Porphyromonas macacae* ^n^	
44	PD3	*Bacteroides pyogenes* ^n^	
45	PD4	*Fusobacterium polymorphum* ^n^	
46	PD3	*Pasteurella canis*	-
47	PD3	*Neisseria animaloris*	Chlor, Amp

* PD, periodontal disease; ** NCBI, National Center for Biotechnology Information; -—resistance not detected; n—susceptibility testing not performed; Amp—ampicillin; Cip—ciprofloxacin; Sxt—sulfamethoxazole-trimethoprim; Doxy—doxycycline; Chlor—chloramphenicol; Col—colistin; Ctx—cefotaxime.

**Table 3 microorganisms-12-01455-t003:** Genes encoding antimicrobial resistance detected in samples of dogs with PD.

Antimicrobial Class	Genes Encoding Resistance
β-lactams	Class D beta-lactamase *OXA-347*
Tetracyclines	Tetracycline resistance ribosomal protection protein *Tet(M)* Tetracycline resistance ribosomal protection protein *Tet(O)* Tetracycline resistance ribosomal protection protein *Tet(32)* Tetracycline resistance ribosomal protection protein *Tet(W)* tetracycline resistance ribosomal protection protein *Tet(Q*)
Macrolides and lincosamides	23S rRNA (adenine(2058)-N(6))-methyltransferase *Erm(F*) 23S rRNA (adenine(2058)-N(6))-methyltransferase *Erm(39)* macrolide efflux MFS transporter *Mef(En2)*
Aminoglycosides	Aminoglycoside 6-adenylyltransferase *AadE* Aminoglycoside 6-adenylyltransferase *AadS*
Colistin	Phosphoethanolamine--lipid A transferase *MCR-1.6* Phosphoethanolamine--lipid A transferase *MCR-1.11*
Different classes (lincomycin, clindamycin, dalfopristin, and tiamulin)	ABC-F type ribosomal protection protein *Lsa(C)*

## Data Availability

The original contributions presented in the study are included in the article/Appendix A. Further inquiries can be directed to the corresponding author.

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
