# Peer review of "Microbial Composition of Extracted Dental Alveoli in Dogs with Advanced Periodontitis"

_microorganisms, 2024, doi:10.3390/microorganisms12071455_

Round 1

Reviewer 1 Report

Comments and Suggestions for Authors

With this symptomatology, dental care of any kind should be considered.

That, in my opinion, is the main problem.

One important question, is wet or dry food fed?

Are the teeth brushed?

It would be good if colleagues could comment this.

Author Response

Please find the response in the file attached. 

Reviewer 2 Report

Comments and Suggestions for Authors

Revision of the manuscript entitled “

 Microbial composition of extracted dental alveoli in dogs with 2 advanced periodontitis”

The manuscript Is well written and the study design is simple. From time to time the study is a little bit boring since it is a simple description of the isolated bacteria. It would be interesting if the authors collected information about the previous administration of antibiotics in the enrolled dogs. And if they also provided information about the possible risk of antimicrobial resistance considering the isolated strains. Please consider implementing the paper according to this last suggestion.

Here I provide more detailed suggestions and comments:

Abstract:

L14: what does MGS mean?

In the abstract, the authors did not mention the methods used to isolate and identify the bacteria. They just resume the isolated bacteria and are peremptory saying that some bacteria are for sure the etiological agents of periodontitis. It would be better to be less categorical and to give more nuanced information.

Introduction

L42: eliminate “.” After [12]

L54-55: Tannerella forsythia, and Campylobacter rectus (italics!)

Materials and Methods

2.1. Animals involved in the study: first describe everything, then the reader can understand the schematic representation. The same for Table 1. First the description, then the link to the table.

Did the authors think about exclusion criteria?
*take care about the size font of figure and table captions *

Results

L214: first explanation, then reference to Figure 4.

Discussion

L249: remove the first sentence since it is a repetition of the introduction

Author Response

Please find the response in the file attached

Reviewer 3 Report

Comments and Suggestions for Authors

Microbial composition of extracted dental alveoli in dogs with advanced periodontitis

 In this study authors wanted to investigate the microbiota using meta-genomic sequencing of extracted tooth alveoli in dogs with severe periodontitis, and to compare microbiological data obtained by MGS with those obtained by traditional plating used for bacteriological diagnosis.

It is an interesting work, although in my point of view the study has some flaws. One of the objectives was not achieved as the comparison between metagenomics and culture cannot be made in this way. In culture, each sample was analyzed on its own, while in metagenomics a pool was made. So there is a bias when combining all the samples. Of course, it gives an average of the present in the different samples, but it cannot be comparable to what is in the culture analyzed separately. Here too, the fact that the authors chose 3 representative colonies adds another factor of distortion.

Another flaw in the study is the sample size, comparing a sample of 30 with another of 6 is not correct. Comparisons cannot be made with statistical strength. I understand that it is not easy to obtain healthy dogs at the periodontium level, but authors should have extended the recruitment time.

I think it will be interesting to get a larger sample of healthy dogs and perform metagenomics for the samples separately. Only then can the conclusions be considered.

Also, as the authors mention, the collection environments between healthy dogs and those with periodontitis are not the same, and this fact obviously influences the type of microorganisms present, so the statements/conclusions to be drawn must be made carefully.

Author Response

(The authors gave the same response as above.)

Round 2

Reviewer 3 Report

Comments and Suggestions for Authors

Microbial composition of extracted dental alveoli in dogs with advanced periodontitis

I thank the authors for their answers to the questions posed and I consider that the change in the direction and focus of the paper was appropriated.

However, there are still some points to correct/improve:

Page 2 – line 96 – “Since pet owners can’t escape close contact with their dogs,…”

Page 5 – line 168 – “The sample was placed into 2-mL cryogenic tube and stored at −80°C until DNA extraction …”

Page 8 – line 257 – “ Although Pasteurella was the most frequently detected genus from culturable isolates, …”

Page 14 – First paragraph, about microbiota of healthy dogs is unnecessary. To be maintained, the text must be revised (all paragraph). It's confusing. English has to be improved. And it is not clear which studies are being referred to when talking about microorganisms.

Page 14 – line 426 – “According to the other studies, tooth brushing…”

Page 14 – line 438 – “Multiresistant isolates were detected in dogs…” – In the dogs of the study? This is not clear in the results section! Must clarify

Page 15 – line 466 – “…, meaning that bacteriological diagnosis is unable to provide a real-world view of the microbial variety at sites of infection.” “Is” must be replaced by “could be” or “sems to indicate”. This work does not allow us to state definitively that the bacteriological diagnosis is not capable, it can be said that it seems to indicate this, but not to affirm it.

The same to the statement “This can lead to an incorrect bacteriological diagnosis…”. It must be changed to “This could lead to an incorrect bacteriological diagnosis…”.

Page 15 – line 473 – “… therefore treatment and prophylaxis of periodontal disease…”

Comments on the Quality of English Language

A careful review of the entire manuscript is necessary

Author Response

Please find the file attached
